# Shortwave Ultraviolet Persistent Luminescence of Sr_2_MgSi_2_O_7_: Pr^3+^

**DOI:** 10.3390/ma16051776

**Published:** 2023-02-21

**Authors:** Andris Antuzevics, Guna Doke, Guna Krieke, Pavels Rodionovs, Dace Nilova, Jekabs Cirulis, Andris Fedotovs, Uldis Rogulis

**Affiliations:** Institute of Solid State Physics, University of Latvia, 8 Kengaraga str., LV-1063 Riga, Latvia

**Keywords:** melilite, afterglow, UVC, long-lasting luminescence, thermostimulated luminescence (TSL), electron spin resonance (ESR)

## Abstract

Currently, extensive research activities are devoted to developing persistent phosphors which extend beyond the visible range. In some emerging applications, long-lasting emission of high-energy photons is required; however, suitable materials for the shortwave ultraviolet (UV–C) band are extremely limited. This study reports a novel Sr_2_MgSi_2_O_7_ phosphor doped with Pr^3+^ ions, which exhibits UV–C persistent luminescence with maximum intensity at 243 nm. The solubility of Pr^3+^ in the matrix is analysed by X-ray diffraction (XRD) and optimal activator concentration is determined. Optical and structural properties are characterised by photoluminescence (PL), thermally stimulated luminescence (TSL) and electron paramagnetic resonance (EPR) spectroscopy techniques. The obtained results expand the class of UV–C persistent phosphors and provide novel insights into the mechanisms of persistent luminescence.

## 1. Introduction

Materials exhibiting persistent luminescence outside the visible light spectrum have received considerable scientific attention [1,2,3,4,5,6,7,8,9,10,11,12,13,14,15,16,17,18,19]. Because of the constraints of human perception, practical interest was initially limited to visible range persistent phosphors for applications in safety signage, toys, and decorations [20,21,22]. However, the electromagnetic spectrum extends far beyond our visual capabilities and many interesting phenomena are related to infrared (IR) and ultraviolet (UV) spectral ranges. One of the better-known aspects of IR radiation is its ability to penetrate living tissue [23]; therefore, non-invasive bioimaging employing IR-emitting persistent phosphor nanoparticles is a promising and rapidly advancing field of research [1,2,3]. On the other hand, the interaction of higher-energy UV-radiation photons with matter can induce a variety of physical, chemical, and biological effects such as photoionization, breakage of chemical bonds, and fluorescence, etc. Consequently, the viability of UV-persistent phosphors for application in photodynamic therapy, photocatalysis, and sterilization has been tested [4,5,6,7,8,9]. In addition, the aspect of afterglow being invisible, both in IR and UV spectral ranges, is advantageous in itself within several fields such as anticounterfeiting, optical tagging, surveillance, and night vision [8,9,10,11,12,13,14,15,16,17,18,19]. Therefore, the development of novel materials with long-lasting luminescence beyond the visible range is crucial to meet the needs of emerging applications.

The engineering of UV-emitting persistent phosphors involves the selection of an appropriate host compound and activator ion combination [8,9,24]. The UV spectrum commonly is divided into three ranges: UV–A (315–400 nm), UV–B (280–315 nm), and UV–C (200–280 nm). To enable radiative transitions in the required spectral range, the host matrix should possess a sufficiently wide band gap and a crystal structure that can accommodate emission centres. Suitable activators for UV persistent phosphors are currently limited to Ce^3+^ (UV–A) [25], Pr^3+^ (UV–B, UV–C) [5,7,10,15,18,19,26,27,28], Gd^3+^ (UV–B) [11,27], Tb^3+^ (UV–A) [29], Pb^2+^ (UV–A, UV–B) [30,31], and Bi^3+^ (UV–A, UV–B, UV–C) [4,6,12,14,17,27,32,33,34,35,36,37,38,39,40,41,42]. The final crucial aspect of persistent phosphor design is related to point defects serving as trapping centres which store excitation energy by capturing charge carriers. Important information regarding the identity, depth, density, and distribution of the traps must be established to optimize the intensity and duration of persistent luminescence. Targeted actions during sample synthesis are often hindered by the lack or ambiguity of evidence regarding the nature of trapping sites which could be partially resolved by combining luminescence spectroscopy with magnetic resonance-based techniques in defect analysis. Because of the above-mentioned considerations, the current knowledge base on UV–C persistent phosphors is quite limited.

An overview of the reported UV–C persistent phosphors is presented in Table 1. The selection of complex oxides as hosts is motivated by their relatively simple preparation process, high stability, and abundance of intrinsic defects that could serve as charge traps. To achieve UV–C persistent luminescence, the choice of emission centres falls almost exclusively to Pr^3+^. The excited 4f^1^5d^1^ state of Pr^3+^ is lattice-sensitive; therefore, the parity allowed in a 4f^1^5d^1^→4f^2^ transition can be tuned over a sufficiently broad UV spectral range [9]. To achieve persistent luminescence from 260 nm, broadly available excitation sources such as 254 nm UV lamps can be used; however, efficient persistent luminescence peaking at lower wavelengths typically requires the use of higher energy sources such as X-rays.

Sr_2_MgSi_2_O_7_ is a promising material to be considered as a UV–C persistent phosphor. Numerous studies have focused on the optical properties of Sr_2_MgSi_2_O_7_ systems doped with rare earth (RE) ions [43,44,45,46,47,48,49,50,51,52,53]; moreover, Sr_2_MgSi_2_O_7_: Eu^2+^, Dy^3+^ is one of the most efficient blue-emitting persistent phosphors [48,49,50,51,52]. Persistent luminescence is enabled by appropriate lattice defects serving as trapping centres; therefore, the nature of defect types in the host and the effect of RE ion substitution on their activation energies have been the subjects of considerable scientific interest [48,49,50,51,52,54,55,56,57]. The sufficiently wide band gap E_g_ = 7.1 eV of Sr_2_MgSi_2_O_7_ [56,57] is a crucial aspect enabling emission in the UV range. UV photoluminescence has been achieved in Gd^3+^ [53] and Pb^2+^ [58] doped Sr_2_MgSi_2_O_7_; however, there are no reports exploring the UV–C persistent luminescence capabilities of the material.

This study reports a novel Sr_2_MgSi_2_O_7_: Pr^3+^ phosphor exhibiting UV–C persistent luminescence after excitation with UV radiation or X-rays. The optimal activator concentration was determined for efficient persistent luminescence. Photoluminescence (PL), thermally stimulated luminescence (TSL), and electron paramagnetic resonance (EPR) techniques were applied to establish the mechanism of persistent luminescence in the material.

## 2. Materials and Methods

Single-phase tetragonal Sr_2_MgSi_2_O_7_ samples doped with 0–3% Pr^3+^ were prepared using solid-state synthesis. Pr^3+^ was introduced in the host by replacing Sr^2+^ and the general composition of the investigated material was Sr_2-x_Pr_x_MgSi_2_O_7_ (x = 0–0.06). High-purity SrCO_3_ (99.994%, Alfa Aesar, Thermo Fisher GmbH, Kandel, Germany), MgO (99.99%, Alfa Aesar), SiO_2_ (99.995%, Alfa Aesar), and Pr_6_O_11_ (99.996%, Alfa Aesar) were used as precursors. Appropriate amounts of precursors were thoroughly mixed in an agate mortar and pressed into 13 mm pellets using a uniaxial hydraulic press from Specac. The pellets were placed on a Pt foil and heat-treated at 1300 °C for 24 h using 5 °C/min heating and cooling rates. The relatively long heat treatment was required to ensure the completion of the solid-state reaction.

The crystal structure of the prepared samples was analyzed using X-ray diffraction (XRD) (Rigaku MiniFlex 600, Rigaku, Tokyo, Japan) with a λ = 1.5406 Å Cu Kα radiation source operating at 40 kV and 15 mA. The phase composition was calculated using Rietveld refinement with Profex software (version 4.1.0) [59].

Photoluminescence (PL) emission and excitation spectra were measured using a spectrometer FLS1000 from Edinburgh Instruments (Livingston, UK) with a Xe lamp as an excitation source. All luminescence and excitation spectra measurements were corrected for the spectral sensitivity of the equipment.

Thermally stimulated luminescence (TSL) curves were measured using a Lexsyg research fully-automated TL/OSL reader from Freiberg Instruments GmbH (Saxony, Germany). As the irradiation source, an X-ray tube VF-50J/S (40 kV, 0.5 mA) was used. TSL curves were recorded using a photomultiplier Hamamatsu R13456. The system was operated at a linear heating rate of 1 °C/s at a temperature range between room temperature and 300 °C. The same system was used to measure isothermal afterglow decay kinetics at 25 °C and TSL spectra. For TSL spectra, the TL/OSL reader was coupled with an Andor SR-303i-B spectrometer with a DV-420A-BU2 CCD camera. For UV (232 nm) excitation, a Nd: YAG Q-switched laser NT342/3UV (pulse duration—4 ns) from Ekspla was used. A UV 310 nm shortpass filter XUV0310 from Asahi Spectra was used to detect UV–C emission.

Electron paramagnetic resonance (EPR) investigations at X (9.363 GHz) and Q (33.92 GHz) microwave frequency bands were performed on the Bruker ELEXSYS-II E500 CW-EPR system (Bruker Biospin, Rheinstetten, Germany) equipped with an Oxford Instruments liquid helium flow cryostat. X-band EPR spectra were detected at 80 K, 10 mW microwave power, 0.1 mT magnetic field modulation amplitude, and 100 kHz modulation frequency. For Q-band measurements, the following parameter values were used: 90 K, 0.26 mW, 0.4 mT, and 100 kHz. Prior to measurements, the sample was irradiated for 10 min at room temperature using an X-ray tube operated at 50 kV, 10 mA. Afterwards, stepwise isochronal (10 min at each step) annealing of the sample was carried out in a custom-built furnace with an estimated temperature uncertainty of ±10 °C. EasySpin toolbox for MATLAB software (version R2020a) [60] was used for EPR spectra simulations.

## 3. Results and Discussion

XRD analysis of Sr_2_MgSi_2_O_7_ samples doped with 0–3% Pr^3+^ is presented in Figure 1. The XRD peak positions and relative intensities are consistent with the reference pattern of tetragonal Sr_2_MgSi_2_O_7_ (PDF 01-079-8255 [61]). No additional peaks are detected for samples doped with up to 1% Pr^3+^ suggesting the incorporation of Pr^3+^ within the Sr_2_MgSi_2_O_7_ lattice. In the sample with the highest Pr^3+^ content, additional peaks associated with Pr^3+^-rich hexagonal strontium silicate SrPr_4_(SiO_4_)_3_O (PDF 04-016-8403 [62]) can be detected indicating that Pr^3+^ solubility in Sr_2_MgSi_2_O_7_ is limited. SrPr_4_(SiO_4_)_3_O content in the sample was estimated to be 2% using Rietveld refinement. 

Sr_2_MgSi_2_O_7_ possesses a melilite-type structure in which the host elements are distributed over six cationic and anionic sites. There are three distinct cationic positions: Sr position with eight-fold, Mg with four-fold, and Si positions with four-fold coordination [61]. It is expected that Pr^3+^ predominantly incorporates within the Sr^2+^ sites because of similar ionic radii of Sr^2+^ (1.260 Å) and Pr^3+^ (1.126 Å [63]). Such substitution requires charge compensation which is likely achieved by cation vacancies present in the material [54]. For oxygen atoms, there are three types of anionic sites in the crystal [61]. First-principle calculations have demonstrated that oxygen vacancies of any type are ideal electron traps in the Sr_2_MgSi_2_O_7_ structure [55], thus encouraging the exploration of the material’s optical properties.

PL spectra of Sr_2_MgSi_2_O_7_ doped with 0.1–3.0% Pr^3+^ are shown in Figure 2. In PL excitation spectra, a single band at a maximum of 232 nm is detected. This corresponds to the interconfigurational 4f^2^→4f^1^5d^1^ transition of Pr^3+^ which has been previously reported in other silicates [18,64,65]. The excitation of this band results in intense UV–C luminescence with the highest intensity of emission detected for the 0.5% Pr^3+^ sample (see inset of Figure 2a). In samples with higher activator content, a gradual decline of emission intensity is observed most likely because of concentration quenching. The dominant emission peaks are located at 243, 253, 276, and 282 nm, which are associated with transitions from 4f^1^5d^1^ to ^3^H_4_, ^3^H_5_, ^3^H_6_, and ^3^F_2_ states, respectively. The relative intensity of the 4f^1^5d^1^→^3^H_4_ emission is slightly smaller at higher levels of Pr^3+^ content, presumably because of the increased efficiency of reabsorption by the closely located 4f^2^→4f^1^5d^1^ band. In addition, radiative transitions from ^3^P_0_ and ^1^D_2_ can be detected in all samples; however, the efficiency of these transitions is insignificant.

For all samples, a strong afterglow signal is observed when irradiated by UV radiation and X-rays. Persistent luminescence spectra after irradiation with X-rays are shown in Figure 3a. The main spectral features coincide with the PL spectra (Figure 2); i.e., predominantly UV–C luminescence bands originating from Pr^3+^ 4f^1^5d^1^→4f^2^ optical transitions were detected. In addition, a relatively weak luminescence band at around 320–400 nm is detected in the persistent luminescence spectra. The comparison of spectral properties of undoped and doped samples with various activators revealed that this band originates from 5d^1^ → ^2^F_5/2_ and ^2^F_7/2_ transitions of Ce^3+^. Separation of adjacent lanthanides is challenging because of small deviations in chemical properties [66]. As a result, Ce^3+^ is a common impurity in Pr^3+^-containing precursors, including the Pr_6_O_11_ [67] used in the present research. However, the contribution of Ce^3+^ emission in persistent luminescence of Sr_2_MgSi_2_O_7_ is comparatively low. Figure 3b shows that UV–C persistent luminescence is detectable for more than 16 h with the highest intensity of emission observed for the 0.7% Pr^3+^ sample. Surprisingly, persistent luminescence in the UV–C range could also be induced by 232 nm laser radiation corresponding to the 4f^2^→4f^1^5d^1^ transition of Pr^3+^ which suggests that Pr^3+^ could be photoionized during the excitation thus acting as an electron donor. The corresponding spectrum and decay kinetics of the 0.7% Pr^3+^ sample are presented in Figure 3c. The results show identical spectral features and highly comparable decay behaviour to the X-ray-irradiated sample. 

TSL analysis of the Sr_2_MgSi_2_O_7_: Pr^3+^ samples after X-irradiation was performed to characterize trap properties. TSL glow curves of the samples are shown in Figure 4a. In general, individual TSL glow peaks represent the number of different types of traps; the intensity and position of each peak correlate with the filled trap density and depth, respectively [68]. The investigated samples may be characterized by a relatively large number of traps with varying trap depth values. The traps can be tentatively divided into shallow traps (glow peaks up to 125 °C) and deep traps (glow peaks above 125 °C), with the deep traps being more prominent in all samples. The highest integral intensity of shallow traps represented by the low-temperature glow peaks was detected for the 0.7% Pr^3+^ sample (inset of Figure 4b). As afterglow at room temperature is mainly caused by the detrapping of shallow traps, the result is consistent with persistent luminescence decay kinetics (Figure 3b). 

A wavelength-resolved TSL contour plot of the 0.7% Pr^3+^ sample shown in Figure 4c yields intriguing results. The origin of TSL emission is strongly dependent on temperature. If the heating temperature does not exceed 100 °C, TSL spectra match persistent luminescence spectra with the dominant luminescence bands corresponding to Pr^3+^ UV–C emission. When temperature exceeds 125 °C, UV–C emission is almost completely quenched, and intense Ce^3+^-related emission emerges. Therefore, it can be concluded that Pr^3+^-related UV–C persistent luminescence is linked to the detrapping of the shallow traps. These results correlate with the persistent luminescence spectra shown in Figure 3a and the TSL glow curves presented in Figure 4b where UV–C emission is predominant.

For a more thorough analysis of trap properties, the partial thermal cleaning (T_max_–T_stop_) experiment, together with initial rise analysis (IRA), were performed for the 0.7% Pr^3+^ sample. In Figure 5a, each TSL glow curve has been measured after preheating the irradiated sample to different temperatures (T_stop_), and subsequently cooling it to room temperature. Developed by McKeever in 1980 [69], the T_max_–T_stop_ analysis is a widely acknowledged experimental method for determining the nature of traps. Constant T_stop_-independent glow peak maximum temperatures (T_max_) are expected for discrete trapping sites. Obviously, this is not the case for the investigated sample, where a gradual shift of T_max_ to higher values with the increase of T_stop_ is observed. Such behaviour strongly indicates that the activation energies of traps in Sr_2_MgSi_2_O_7_ are continuously distributed [70,71].

To determine trap depth values *E_a_*, all glow curves were analyzed by applying IRA. This method assumes that the initial low-temperature side of the TSL peak will follow the Arrhenius equation [72]:(1)$$I\left(T\right)=C\cdot \exp\left(-\frac{E_{a}}{k_{B}T}\right)$$
where $I\left(T\right)$—intensity as a function of temperature; C—a constant that includes a frequency factor (assumed to be independent of temperature); and T—temperature. According to Equation (1), the initial rise part of the glow peak is represented by a straight line with a slope of −$E_{a}$ if $\ln\left(I\right)$ is plotted as a function of $\frac{1}{k_{B}T}$. Analysis of selected glow curves is shown in Figure 5b with linear fits indicated by the blue dashed lines. The obtained trap depth values cover the energy region between 0.68–0.95 eV.

Additionally, the method described in ref. [73] was used to estimate the density distribution of filled traps. It is based on a calculation of the difference between integrated intensities of two consecutive TSL glow curves from the T_max_–T_stop_ experiment. The integrated intensity is directly related to the total number of filled traps; therefore, the difference in integrated intensities between two glow peaks from the T_max_–T_stop_ experiment will correspond to the number of traps emptied with an increased T_stop_ value. As seen in Figure 5c, the shallow traps may be characterized by at least three types of trapping levels, where each type can be characterized by a quasi-continuous distribution.

Analogous analysis was carried out for the deep traps, for which a quasi-continuous distribution of trapping levels exists as well. The calculated trap depth values were determined to lie between 1.25–1.45 eV, whereas maximum trap densities were observed at 196, 278, and 376 °C. These results, however, do not provide insight into the origin of trapping sites; therefore, additional spectroscopic measurements were performed.

The response of Sr_2_MgSi_2_O_7_: Pr^3+^ to optical stimulation was tested by analysing TSL glow curves after bleaching with either 458, 590, or 850 nm light-emitting diodes. A minor diminishing effect on TSL intensity was observed with the bleaching effect being more pronounced for 458 nm light. Additional experiments performed along the lines described in refs. [74,75] should be considered to characterize the effects of photostimulation in the material.

EPR spectroscopy was applied to characterize radiation-induced centre formation and stability in Sr_2_MgSi_2_O_7_; the results of EPR analysis are summarized in Figure 6. Figure 6a demonstrates that “EPR-active” centres (electron spin *S* ≠ 0) are generated after irradiation of the sample with X-rays. Resonance signals are observed in 325–350 mT range, which for the experimental X-band microwave frequency corresponds to *g*-factor values of 2.06–1.91. Their relative intensities suggest that the experimental spectrum is composed of several overlapping signals. To verify this assumption, EPR spectra were recorded after annealing the sample at different temperatures (Figure 6b). EPR signals annihilate in several stages, providing experimental evidence for the presence of at least three paramagnetic centres which are labelled “Centre I-III” in the order of ascending stability.

Simulations of the EPR data recorded at two microwave frequencies were performed to determine spin–Hamiltonian (SH) parameters of the individual signals (Figure 6c,d). A multifrequency approach in EPR analysis is particularly useful to ensure a precise and unambiguous analysis of highly anisotropic signals [76]. The following SH was used in the simulations:(2)$$H=g\mu_{B}BS+SAI$$

In Equation (2), g is the *g*-factor; $\mu_{B}$—the Bohr magneton; B—external magnetic field; S—electron spin operator; A—the hyperfine (HF) coupling tensor; I—nuclear spin operator [77,78]. All paramagnetic centres were identified as S = 1/2 systems; for Centre I, weak HF interaction with Mg nuclei (10% abundant ^25^Mg isotope with I = 5/2) was partially resolved. A summary of the fitted SH parameter values is provided in Table 2.

The nature of X-radiation-induced paramagnetic centres can be discussed on the basis of simulation results. In oxide hosts, trapped electron centres are typically characterized by g < g_e_ = 2.0023, whereas trapped-hole centres exhibit g > g_e_ [79,80,81]. The determined SH parameter values of Centre I imply that it is a single trapped hole in the vicinity of a Mg nucleus. Oxygen ions are common hole traps in oxides resulting in the formation of *S* = 1/2 O^−^ ions [82,83,84]. Of the three unique oxygen sites in the Sr_2_MgSi_2_O_7_ structure, only one is coordinated by a single Mg ion (O3 in ref. [61]). Moreover, the Mg–O distance and the magnitude of HF coupling are comparable to that of similar centres in MgO [84]. Therefore, it can be proposed that trapped holes at the O3 site give rise to the EPR signal of Centre I. Not much can be inferred about the other EPR signals without discernible spectra-HF structure. The determined g values suggest that Centre II is most likely comprised of single electrons trapped at vacant oxygen sites and that Centre III could be another trapped-hole centre variation.

All centres are relatively stable at 50–100 °C; however, at higher temperatures, each paramagnetic centre exhibits distinct thermal properties (Figure 7). Annihilation of Centre I occurs rapidly at 100–150 °C, while Centre II anneals more gradually at 100–250 °C. The asynchronous decay of trapped holes (Centre I) and electrons (Centre II) suggests that several recombination pathways exist for the X-ray induced paramagnetic centres. Centre III signal evolution with annealing temperature could imply that, besides thermally stimulated recombination, re-trapping processes of charge carriers also occur in the material. However, large uncertainties due to the spectral overlap with the dominant Centre I signal inhibit definitive conclusions.

Several points can be made regarding the mechanism of persistent luminescence in Sr_2_MgSi_2_O_7_: Pr^3+^. The band gap of the material is 7.1 eV [56,57] (175 nm); therefore, the location of the 4f^1^5d^1^ emitting state of Pr^3+^ could be expected to be close to the conduction band. Excitation of this band results in persistent luminescence (Figure 3c) which suggests that Pr^3+^ can be partly photoionized and could act as a charge trap centre (electron donor). UV–C persistent luminescence of Pr^3+^ results from the recombination of shallow charge traps with activation energies within a 0.68–0.95 eV range (Figure 5). Only a slight correlation between the low-temperature TSL data and paramagnetic trapped-hole centre stability is observed (Centre I in Figure 7), suggesting that UV–C persistent luminescence is mainly related to the gradual liberation of non-paramagnetic defects such as F centres with two trapped electrons. Besides Pr^3+^ UV–C emission, TSL emission from trace impurity Ce^3+^ ions is present at higher temperatures (Figure 3c). It is associated with thermally assisted recombination of relatively stable defects which could be related to the paramagnetic defects identified in Figure 6. However, these defects are stable at room temperature (Figure 7) and do not contribute to the persistent luminescence of Sr_2_MgSi_2_O_7_: Pr^3+^. As a result, persistent luminescence can be tuned by temperature (Figure 4c), which is a promising aspect for anticounterfeiting applications.

## 4. Conclusions

A novel Sr_2_MgSi_2_O_7_ phosphor doped with Pr^3+^ ions exhibiting UV–C persistent luminescence with an emission maximum of 243 nm has been successfully developed. It was determined that 0.7% was the optimal dopant content for achieving the highest intensity of UV–C persistent luminescence. The persistent luminescence is detected after excitation with both UV radiation and X-rays which is promising for practical applications.

UV–C persistent luminescence of Sr_2_MgSi_2_O_7_: Pr^3+^ results from recombination processes of shallow traps with activation energies within a 0.68–0.95 eV range. Excitation radiation generates three spin *S* = 1/2 centres which can be associated with singly trapped-hole and electron centres in the material. However, the relatively high stability of the detected paramagnetic centres implies that UV–C persistent luminescence is caused by other defects.

## Figures and Tables

**Figure 1 materials-16-01776-f001:**
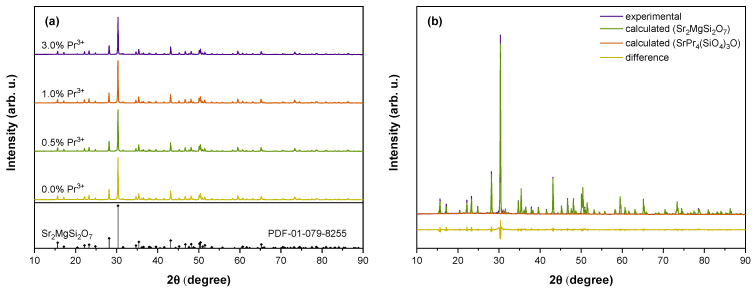
(**a**) XRD patterns of Sr_2_MgSi_2_O_7_ samples doped with 0–3% Pr^3+^; (**b**) Rietveld refinement data of the 3.0% Pr^3+^ sample.

**Figure 2 materials-16-01776-f002:**
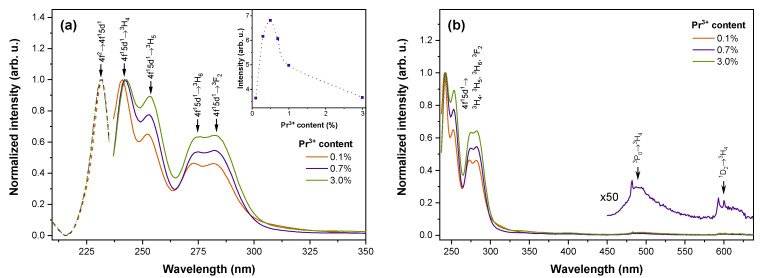
(**a**) PL excitation spectra monitoring 243 nm emission (dashed lines) and PL spectra in the UV range excited with 232 nm (solid lines); inset: integral UV-C emission intensity dependence on Pr^3+^ content; (**b**) PL spectra in UV and visible ranges excited with 232 nm of Sr_2_MgSi_2_O_7_ doped with 0.1–3.0% Pr^3+^.

**Figure 3 materials-16-01776-f003:**
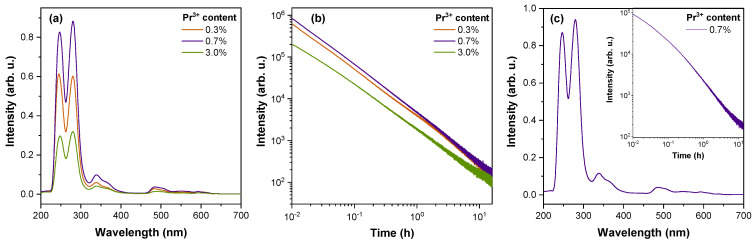
(**a**) Persistent luminescence spectra and (**b**) UV-C emission decay kinetics of Sr_2_MgSi_2_O_7_ samples doped with 0.3–3.0% Pr^3+^ after irradiation for 3 min with X-rays at room temperature; (**c**) persistent luminescence spectrum of the 0.7% Pr^3+^ sample irradiated for 10 min with 232 nm; inset: UV-C persistent luminescence decay kinetics of the same sample.

**Figure 4 materials-16-01776-f004:**
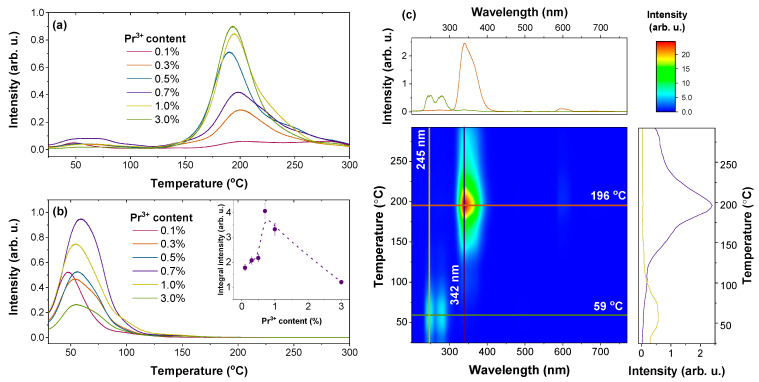
TSL glow curves of (**a**) full emission and (**b**) filtered UV-C emission of Sr_2_MgSi_2_O_7_ samples doped with 0.1–3.0% Pr^3+^ after 30 s irradiation with X-rays; inset: integral TSL intensity dependence on Pr^3+^ concentration; (**c**) wavelength-resolved TSL contour plot of Sr_2_MgSi_2_O_7_ doped with 0.7% Pr^3+^.

**Figure 5 materials-16-01776-f005:**
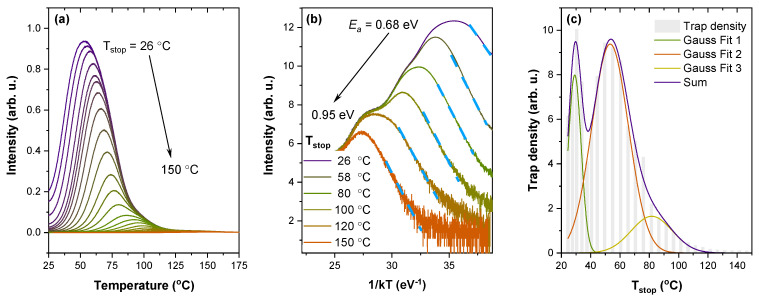
(**a**) TSL glow curves of the 0.7% Pr^3+^ sample measured after preheating to T_stop_ from 26 to 150 °C; (**b**) IRA of selected data obtained from the T_max_–T_stop_ experiment; (**c**) the calculated trap density distribution.

**Figure 6 materials-16-01776-f006:**
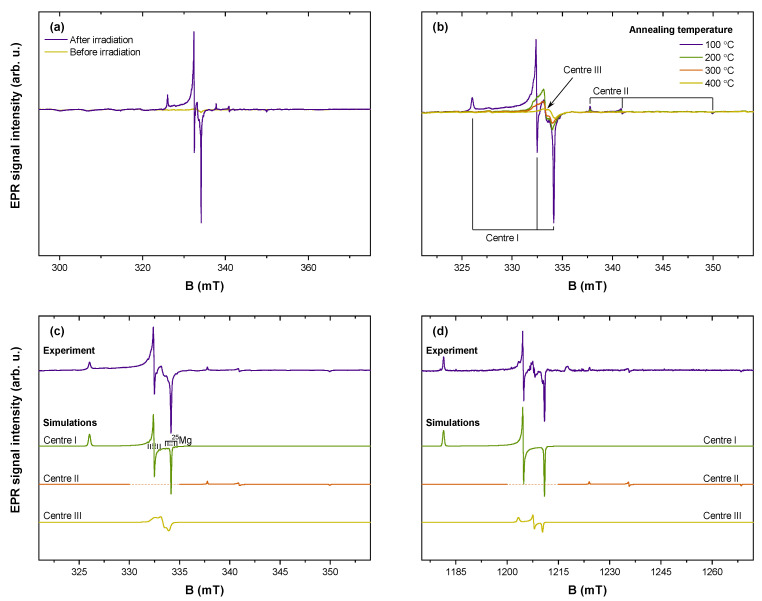
EPR analysis of the 0.7% Pr^3+^ sample: (**a**) experimental spectra before and after irradiation with X-rays; (**b**) spectra evolution with sample annealing temperature; simulations of individual signals contributing to EPR spectra detected at (**c**) X and (**d**) Q microwave frequency bands.

**Figure 7 materials-16-01776-f007:**
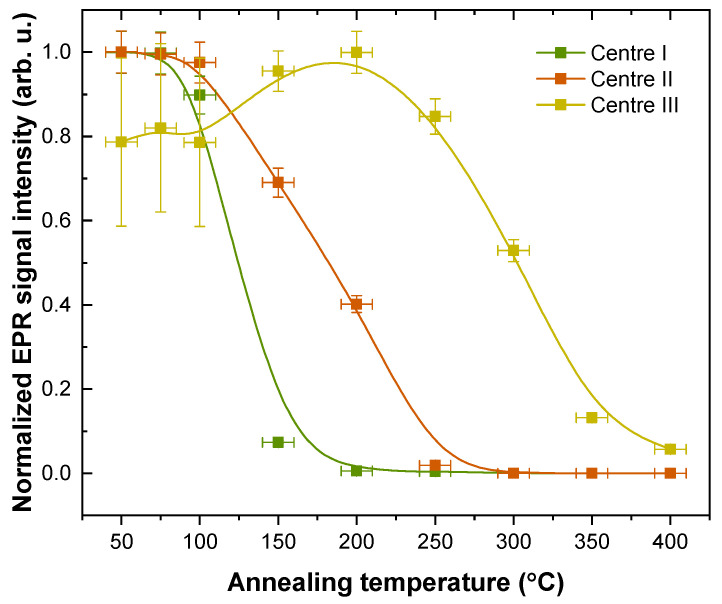
Annealing kinetics of the individual EPR signals.

**Table 1 materials-16-01776-t001:** An overview of UV–C persistent phosphors.

Host	Emission Centre	Excitation	Maximum of Persistent Luminescence, nm	Reference
Lu_2_SiO_5_	Pr^3+^	UV (254 nm)	270	[10,18]
Ca_2_Al_2_SiO_7_	Pr^3+^	UV (254 nm)	268	[10]
Ca_3_Al_2_Si_3_O_12_	Pr^3+^	UV (254 nm)	267	[10]
LiYSiO_4_	Pr^3+^	UV (254 nm)	267	[10]
(Ca_1.5_Y_1.5_)(Al_3.5_Si_1.5_)O_12_	Pr^3+^	UV (254 nm)	266	[15]
Sr_3_Y_2_Si_6_O_18_	Pr^3+^	UV (254 nm)	265	[10]
Li_2_CaGeO_4_	Pr^3+^	UV (254 nm) X-rays	252	[7]
Cs_2_NaYF_6_	Pr^3+^	X-rays	250	[5]
Sr_2_MgSi_2_O_7_	Pr^3+^	X-rays UV (232 nm)	243	This work
YPO_4_	Bi^3+^	X-rays	240	[37]
LaPO_4_	Pr^3+^	X-rays	231	[26]

**Table 2 materials-16-01776-t002:** SH parameters of X-ray-induced paramagnetic centres in Sr_2_MgSi_2_O_7_: 0.7% Pr^3+^. Δg _i_ = 0.0005; ΔA _i_ = 0.5 MHz.

	$g_{1}$	$g_{2}$	$g_{3}$	$A_{1},\mathrm{MHz}$	$A_{2},\mathrm{MHz}$	$A_{3},\mathrm{MHz}$
Centre I	2.0516	2.0120	2.0017		6.6	6.4
Centre II	1.9803	1.9619	1.9111			
Centre III	2.0132	2.0068	2.0028			

## Data Availability

The data presented in this study are available in article.

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
