# Peer review of "Shortwave Ultraviolet Persistent Luminescence of Sr2MgSi2O7: Pr3+"

_materials, 2023, doi:10.3390/ma16051776_

Round 1

Reviewer 1 Report

Manuscript Number: Materials 203019

Although this manuscript explores a novel work related to persistent luminescence, but after analyzing the grammar, formal analysis and figure qualities, it is concluded that the present manuscript cannot be accepted in its present form.

Comment 1:

If the Pr3+ ion replaces the Sr2+ ion in the host matrix, then what about the charge neutrality of the phosphor? Please clarify that, what about the extra +1 charge imbalance?

Comment 2:

The excitation and emission spectra provided in figure 2 is very unclear. The authors are suggested to provide a clear figure? Also, mention the emission wavelength at which the excitation spectra have been recorded.

Comment 3:

The figure qualities and resolutions are very poor. The authors are suggested to improve them.

Comment 4:

The authors are saying that the emission band around 320-400 nm is originating from Ce3+, which is a trace impurity in Pr6O11. This is statement is not justified at all. Presence of any other rare-earth means the sample is contaminated. How can the authors perform the analysis of a contaminated sample?

Reviewer 2 Report

The paper  titled Shortwave ultraviolet persistent luminescence of Sr2MgSi2O7:Pr3+ presents a meticulous study of an interesting material. I found no objections nor I have suggestions for the authors. I suggest accepting the paper as is.

Author Response

Thank you for the appreciation of our study!

Reviewer 3 Report

The manuscript details the novel emission of Pr3+ in Sr2MgSi2O7, the manuscript can be published however requires major revisions.

1- The authors have discussed about the phase purity evaluation through Reitveld Refinements. However no data related to to Reitveld refinements is present in the manuscript. Despite discussing it in "Materials and Methods" section and then in the "Results and discussions". It seems a noticeable lack of concentration during the preparation of manuscript. 

2- The authors should discuss more regarding the luminescence characteristics and presence of Ce3+T

Reviewer 4 Report

Andris Antuzevics et al. have successfully synthesized a novel Sr2MgSi2O7 phosphor doped with Pr3+ ions exhibiting UV-C persistent luminescence with emission maximum located at 243 nm. UV-C persistent luminescence of Sr2MgSi2O7: Pr3+ results from recombination processes of shallow traps with activation energies within 0.68-0.95 eV range. Excitation radiation generates three spin S = 1/2 centres, which can be associated with singly trapped hole and electron centres in the material. A detailed description of the mechanism leading to UV-C persistent luminescence is provided. The manuscript is also thorough and well written. It is worthy of publication though below is the questions that should be considered.

Question 1: Thermally stimulated luminescence (TSL) can be obtained in Fig 5a. Can the optical stimulated luminescence be obtained? Some references (such as Mater. Today Phys. 2022, 27, 100765; Sci. Rep. 2013, 3, 1554) should be cited to explain the difference between optical stimulated luminescence and photoluminescence.

Question 2: The variable temperature afterglow spectrum should be measured to obtain the best afterglow fluorescence to meet the application.
